# The Genetic Diversity and Dysfunctionality of Catalase Associated with a Worse Outcome in Crohn’s Disease

**DOI:** 10.3390/ijms232415881

**Published:** 2022-12-14

**Authors:** Marisa Iborra, Inés Moret, Enrique Busó, José Luis García-Giménez, Elena Ricart, Javier P. Gisbert, Eduard Cabré, Maria Esteve, Lucía Márquez-Mosquera, Esther García-Planella, Jordi Guardiola, Federico V. Pallardó, Carolina Serena, Francisco Algaba-Chueca, Eugeni Domenech, Pilar Nos, Belén Beltrán

**Affiliations:** 1Gastroenterology Department, La Fe University and Polytechnic Hospital, 46026 Valencia, Spain; 2Medical Research Institute Hospital La Fe (IIS La Fe), 46026 Valencia, Spain; 3Central Unit for Research in Medicine (UCIM), Faculty of Medicine and Dentistry, University of Valencia, 46010 Valencia, Spain; 4INCLIVA Biomedical Research Institute, Spanish Institute of Health Carlos III, Department of Physiology, Faculty of Medicine and Dentistry, University of Valencia, Center for Biomedical Research Network on Rare Diseases (CIBERER), 46010 Valencia, Spain; 5Inflammatory Bowel Disease Unit, Gastroenterology Department, Hospital Clìnic de Barcelona, CIBEREHD, IDIBAPS, 08036 Barcelona, Spain; 6Gastroenterology Department, Hospital Universitario de La Princesa, Instituto de Investigación Sanitaria Princesa (IIS-Princesa), Universidad Autónoma de Madrid (UAM), CIBEREHD, 28006 Madrid, Spain; 7Gastroenterology Department, Hospital Germans Trias i Pujol, CIBEREHD, 08916 Badalona, Spain; 8Gastroenterology Department, Hospital Universitari Mutua de Terrassa, CIBEREHD, 08221 Barcelona, Spain; 9Servei de Digestiu, Hospital del Mar, Barcelona, IMIM (Hospital del Mar Medical Research Institute), 08003 Barcelona, Spain; 10Gastroenterology Department, Hospital de la Santa Creu i Sant Pau, 08025 Barcelona, Spain; 11Gastroenterology Department, Hospital Universitari de Bellvitge, Hospital de Llobregat-Barcelona, 08901 Barcelona, Spain; 12Institut d’Investigació Sanitària Pere Virgili, Hospital Universitari Joan XXIII, 43007 Tarragona, Spain; 13Medicine and Surgery Department, University Rovira i Virgili, 43201 Reus, Spain; 14Hospital Vithas Virgen del Consuelo, 46007 Valencia, Spain

**Keywords:** catalase, inflammatory bowel disease, Crohn’s disease, oxidative stress, antioxidant genes

## Abstract

Chronic gut inflammation in Crohn’s disease (CD) is associated with an increase in oxidative stress and an imbalance of antioxidant enzymes. We have previously shown that catalase (CAT) activity is permanently inhibited by CD. The purpose of the study was to determine whether there is any relationship between the single nucleotide polymorphisms (SNPs) in the CAT enzyme and the potential risk of CD associated with high levels of oxidative stress. Additionally, we used protein and regulation analyses to determine what causes long-term CAT inhibition in peripheral white mononuclear cells (PWMCs) in both active and inactive CD. We first used a retrospective cohort of 598 patients with CD and 625 age-matched healthy controls (ENEIDA registry) for the genotype analysis. A second human cohort was used to study the functional and regulatory mechanisms of CAT in CD. We isolated PWMCs from CD patients at the onset of the disease (naïve CD patients). In the genotype-association SNP analysis, the CAT SNPs rs1001179, rs475043, and rs525938 showed a significant association with CD (*p* < 0.001). Smoking CD patients with the *CAT* SNP rs475043 A/G genotype had significantly more often penetrating disease (*p* = 0.009). The gene expression and protein levels of CAT were permanently reduced in the active and inactive CD patients. The inhibition of CAT activity in the PWMCs of the CD patients was related to a low concentration of CAT protein caused by the downregulation of *CAT*-gene transcription. Our study suggests an association between CAT SNPs and the risk of CD that may explain permanent CAT inhibition in CD patients together with low CAT gene and protein expression.

## 1. Introduction

The accurate pathogenesis of Crohn’s disease (CD) has been a hot topic of debate, and several aspects of this disease remain unclear. The formation of reactive oxygen species (ROS) and the consequent oxidative stress generated have surfaced as potential etiological factors as well as a pathway of tissue damage in CD [1,2,3,4]. Moreover, ROS also appear to participate in the regulation of processes, such as the cell cycle and apoptosis [5,6], and defective apoptosis of activated lamina propria T lymphocytes has been proposed as a key pathogenic mechanism of intestinal mucosa damage and chronic inflammation in CD [7].

ROS are removed by antioxidant enzymes (AOEs), such as superoxide dismutases (SOD), catalase (CAT), and glutathione peroxidase, which play a key role in the detoxification of superoxide and hydroperoxide, respectively. Those enzymes contribute to the body’s natural defense against oxidative stress through the activation of free-radical scavenging. Besides their role in detoxifying ROS, AOSs are also involved in relevant metabolic and cellular processes. It is noteworthy that CAT is involved in apoptosis [8] and its administration has been used to prevent or decrease the severity of certain intestinal pathologies and inflammatory bowel diseases [9,10]. Our group has previously described the persistent oxidative damage of immune cells in CD patients both in the acute phase of a flare-up (active, aCD) and in the remission phase of the disease (inactive, iCD) and the permanent inhibition of CAT expression and activity [11,12]. Insufficient levels of CAT can lead to the suppression of autophagy-dependent cell death [13,14], and the association between impaired autophagy and the pathogenesis of CD has been reported [15]. On the other hand, *CAT*-gene mutations have been linked to several autoimmune diseases [16] and can affect gastrointestinal mucosa [17]. Therefore, the genetic variants of *CAT* might be implicated in the pathogenesis of CD.

The aim of this study was to determine whether single nucleotide polymorphisms (SNPs) in the *CAT*-gene were associated with the risk of CD in the Spanish population. Moreover, we investigated the specific causes for the permanent inhibition of CAT in the peripheral blood mononuclear cells of active and inactive CD patients at both the pre- and post-transcriptional levels.

## 2. Results

### 2.1. SNPs in the Catalase Gene Are Associated with the Risk of Crohn’s Disease in a Spanish Population

Clinical and demographic data from the discovery cohort are shown in Table 1. No significant differences in age and gender were found between the CD patients and the control group. A total of 24% of the CD patients had perianal disease. Of note, 82% of CD patients were under immunosuppressors therapy, and 62% were under biologic therapy (mainly anti-TNF drugs). Approximately half of the population had never smoked, and the other half were smokers or exsmokers. The *CAT* SNPs rs1001179, rs475043, and rs525938 showed a significant association with CD (*p* < 0.001) with recessive, codominant, and log-additive models, respectively. Multivariate analysis using a backward logistic regression model showed that the smoking CD patients with the *CAT* SNP rs475043 A/G genotype had significantly more often penetrating disease (*p* = 0.009) (Table 2). The genotype association analysis of the *CAT* SNPs with the clinical variables is shown in Table 2. None of the analyzed SNPs was correlated with age and sex. Of note, we found a significant difference in 4 out of the 14 *CAT*-gene SNPs evaluated (rs7943316, rs475043, rs769217, and rs525938) between the CD patients and the control group (Table 3). Some of these four polymorphisms have been previously implicated with the pathogenesis of inflammatory bowel disease (IBD) [16,18,19].

### 2.2. Catalase Expression Is Permanently Inhibited in Crohn’s Disease Patients

Clinical and demographic data from the confirmatory cohort is shown in Table 4. We assume SNP diversity and distributions will be similar to those observed in the discovery cohort. CRP, fibrinogen, and ESR mean values in active CD were substantially greater than those in inactive CD. All CD patients belonged to the A2 of the Montreal classification (age of diagnosis 17–40 years); most CD patients had ileal localization (L1), without perianal disease, and with inflammatory behavior (B1). The maintenance treatments in which the CD patients achieved remission were immunosuppressive agents (azathioprine) for five patients (two of them as recurrence prevention for previous surgical resection), biological therapy (adalimumab) in one patient, double immunotherapy (azathioprine and infliximab) in two patients, mesalazine in one patient, and one patient was without treatment.

A permanent reduction in CAT expression levels in both active and inactive CD patients was confirmed by qPCR (Figure 1A) and Western blotting (Figure 1B), according to the results obtained from the previously published enzymatic assay [11]. No significant differences in CAT expression were found between active and inactive CD. Interestingly, CAT expression was diminished in inactive CD subjects, indicating that the remission of the disease does not increase *CAT* gene expression. In agreement, densitometric analysis of the WB assays revealed that CAT levels were significantly lower in the active CD group compared to the healthy subjects (Figure 1B). The CAT expression values were normalized with ACTB-6, GAPDH-6, and HMBS-7, and those from the protein levels were normalized with α-actin (which we routinely used).

Finally, we analyzed the protein levels of Protein Phosphatase 2A (PP2A) and Protein Kinase C zeta (PKCζ), two important CAT regulators. While PP2A has been reported to decrease CAT activity, the in vitro studies indicated that PKCζ is capable of increasing it [20]. In concordance, we found a significant increase of PP2A in the active CD patients compared with the control subjects, but surprisingly, those patients also displayed increased PKCζ levels (Figure 1C). Of note, when patients achieved remission, the levels of PKCζ and PP2A returned to values more similar to the controls. However, no significant differences were found between the active and inactive CD groups. These results might indicate that CD displays a dysregulation in these enzymes.

## 3. Discussion

Previously published data regarding the status of antioxidants in CD patients has revealed an imbalance in AOE concentrations and an increase in oxidative stress as potential triggering factors in CD [1,21,22,23,24]. We have previously demonstrated that oxidative stress in CD depends on the increased production of H_2_O_2_. Its detoxification must be independent of CAT activity because this activity is permanently inhibited [11]. On the other hand, previous studies have described that CAT regulation can occur both at transcriptional (mutations or single nucleotide polymorphisms) and post-transcriptional levels [16,20,25,26,27]. TNF-α, the key cytokine involved in CD pathogenesis and treatment, is an important factor with the ability to modify CAT activity [28] and expression [29]. The exposure of rat liver cells to TNF-α induces the inhibition of CAT activity [28] and a down-regulation of CAT mRNA levels [29].

Previous studies have investigated CAT status in IBD patients, and prior works suggest this antioxidant defense is not impaired. No differences were observed in CAT activity between the controls and the inflamed biopsy samples of ulcerative colitis (UC) and CD patients [30,31,32]. However, the CAT enzyme in peripheral blood has not been extensively researched, and our current knowledge comes from in vitro studies and tumoral cell lines [33,34,35]. This has also been described for similar conditions in systemic lupus erythematosus, another chronic autoimmune inflammatory disease [36,37]. In particular, increased SOD activity and malondialdehyde (MDA) levels, the most frequently used biomarkers of oxidative stress, as well as a decrease in CAT and glutathione peroxidase enzymes in the serum of systemic lupus erythematosus patients, have also been reported [38]. The authors indicated that H_2_O_2_ could not be detoxified due to the decreased activities of detoxifying enzymes but are instead converted into hypochlorous acid (HOCl) and hydroxyl radicals (HO˙) by myeloperoxidase, reflected as an increased serum MDA concentration, which in agreement with our findings [11].

*CAT* mutations have been reported in the literature [16] regarding autoimmune diseases such as diabetes mellitus [39] or vitiligo [40] and arterial hypertension [41]. We investigated several *CAT* SNPs described previously in the literature, selected according to their role in other autoimmune and chronic diseases, as well as some forms of cancers in which there is an inhibition of CAT [27,42,43,44,45,46,47]. The *CAT* SNP rs1001179 (C > T) polymorphism of the CAT gene has been the most investigated in the literature. Indeed, it has been related to the risk of UC [48,49]. In our study, the *CAT* SNP rs1001179 CC genotype was associated more strongly with CD patients without perianal disease. The CC genotype is associated with a significant decrease in CAT expression in comparison with the TT and CT genotypes [43,50]. In contrast, there are studies that describe lower CAT activity in the carriers of the *CAT* TT polymorphic allele [51,52]. CAT levels have been reported to be lower in the serum of patients with systemic lupus erythematosus compared to healthy controls [38], including an association with the CC genotype [37]. However, the role of CAT-262C > T in the development of different diseases remains controversial. It has been associated with diseases in which this gene causes lower CAT activity and an increase in oxidative stress [53,54,55]. In contrast, CAT-262C > T polymorphism can protect against the development of other stress-oxidative-mediated pathologies [56,57,58]. This discrepancy can be explained by the different biological mechanisms that take place in the pathogenesis of these diseases and shows the role of CAT-262C > T polymorphism as a variable factor altering AOE structure. Some studies have searched for the possible association between the 389C > T polymorphism (rs769217) in the *CAT* gene and the risk of vitiligo, while others reported a lack of this relationship [44,59,60]. These findings highlight the genetic and epigenetic influence on the development of autoimmune diseases. *Konings* et al. have investigated other *CAT* SNPs, such as rs12273124, rs475043, rs494024, and rs564250 [46]. Of note, the authors discovered that some *CAT* SNPs had an influence on noise-induced hearing loss [46]. Finally, *Kim* et al. studied the SNPs rs2284365, rs3758730, rs525938, and rs7943316 in the *CAT* gene and their association with the risk of osteonecrosis of femoral heads [45].

We discovered a few CAT mutations that might help to explain the persistent inhibition seen in CD patients. But CAT activity changes are dependent on regulatory mechanisms that are still poorly understood in addition to the mutations [61]. In this work, we analyzed, for the first time in a Spanish Cohort, the status of CAT in the PWMCs of CD patients from the initial stages of the disease prior to any medical treatment which might influence the results obtained (naïve CD patients). Of note, some of the active patients achieved remission. In our study, and according to our previous results in which the activity of CAT was continuously inhibited in CD, we found a decrease in its expression levels and protein concentrations in CD patients linked to genetic variations within the *CAT* gene. Moreover, the persistent inhibition of CAT was maintained in the same patients once they achieved remission. Therefore, although the levels of the two CAT regulators PPA2 and PKCζ were elevated in CD, the exact transcriptional and post-transcriptional mechanisms underlying the inhibition of CAT are not fully understood. In this regard, the studied *CAT* polymorphisms may be related with this inhibition and may directly contribute to the pathogenesis of CD.

The influence of CAT activity on cellular processes, such as apoptosis, has been described [8]. Only tumor cells (resistant to apoptotic stimuli) have persistently shown low levels of CAT protein and activity and high levels of ROS [35]. In this sense, the previous work of our laboratory demonstrated the persistent inhibition of the CAT enzyme in CD patients [11], which was in concordance with the downregulation of CAT expression and protein levels that we observed in this work. Some studies pointed to apoptosis resistance in those patients as a possible pathogenic mechanism [7]. The precise mechanism by which the CAT enzyme regulates apoptosis remains undetermined. Some studies have shown that the CAT enzyme is regulated post-transcriptionally by some protein kinases and phosphatases, as well as by other signaling molecules [20,25]. Yano et al. demonstrated that the increase in PKCζ and the decrease in PPA2 were able to modulate CAT activity in vitro [20]. In our study, we analyzed the protein levels of PKCζ and PPA2, which showed an increase in the active CD patients, while the inactive CD patients displayed levels more similar to the controls. PKCζ, a serine-threonine kinase belonging to the atypical subfamily of PKC proteins, regulates apoptosis via the inactivation of caspase 9 [62] and is directly involved in cell proliferation [63]. PKCζ is activated by oxidative conditions in the cell and is capable of increasing CAT activity constitutively [20,64]. The increase in those CAT regulators in the CD patients showed that some cellular mechanisms try to increase CAT activity in order to balance oxidative stress. PPA2, a serine/threonine phosphatase, acts as an inducer of apoptosis and as a human tumor suppressor [60,65] and has been described to decrease CAT activity [20]. PPA2 inactivation was reported due to cellular oxidative stress, which, in turn, causes the activation of NF-κB and its signaling [66]. Thus, the increase in PPA2 at the onset of CD can be related to a mechanism to counteract the increase in PKCζ and/or to compensate for the inhibition of PPA2 due to the presence of H_2_O_2_. Studies that aim to identify the molecules that regulate the CAT enzyme are necessary to improve our knowledge about the pathogenesis of CD and to promote the development of advanced therapies targeting key molecules that are involved in this disease.

The main strengths of this work are a large number of participants for the SNP analysis, which allowed us to extract solid results and conclusions. Secondly, we analyzed, for the first time, the status of the CAT enzyme in the PWMCs of CD patients from the initial stages of the disease prior to any medical treatment that might have influenced the results obtained. On the other hand, our study has some limitations that warrant discussion. Our results show that CAT is a susceptibility gene for CD. However, other factors, such as ambient or epigenetic factors, may contribute to disease onset. Further population studies are needed to evaluate if this polymorphism can be used in clinical practice as a biomarker for different clinical or serological manifestations in CD.

## 4. Materials and Methods

### 4.1. Study Design

#### 4.1.1. Discovery Cohort (Genetic Study)

A retrospective cohort study was based on data obtained from the ENEIDA registry. This cohort included patients from the Spanish National Inflammatory Bowel Disease Study of Genetic and Environmental Factors database ENEIDA, promoted by GETECCU (Spanish Working Group in Crohn’s Disease and Ulcerative Colitis) [67]. The ENEIDA project was approved by Research Ethic Committees in all participating centers. Written informed consent to be enrolled in ENEIDA registry was obtained from all patients or from the legal guardians of minors. We evaluated a Caucasian Spanish cohort of 598 CD and 625 age-matched healthy controls. Blood samples were obtained from all the participants. Genomic DNA was analyzed for the 13 SNPs of the CAT gene (rs1001179, rs12273124, rs17886155, rs2268064, rs2284365, rs3758730, rs475043, rs494024, rs525938, rs564250, rs704724, rs769217, and rs7943316). Genotyping analysis was performed by the Agena Biosciences MassArray^®®^ platform. Males and females aged ≥18 years with an established diagnosis of CD, according to the standard criteria of the European Crohn’s and Colitis guidelines^15^, were included. Of note, none of those patients had received any specific treatment (naïve CD patients).

#### 4.1.2. Confirmatory Cohort (Functional and Regulatory Studies of CAT Gene)

Prospective cohort study. The levels of CAT regulator protein kinase C zeta (PKCζ) and protein phosphatase 2A (PP2A) were analyzed in 18 healthy volunteers (controls) and 20 CD patients at the onset of disease (active CD), diagnosed according to Leonard-Jones criteria [68] prior to any specific treatment. Patients were characterized according to the Montreal classification [69], and disease activity was scored based on the Harvey–Bradshaw index [70]. A total of 10 patients achieved clinical, analytical, and morphological remission (iCD) after specific treatment (5-aminosalycilates, corticosteroids, immunosuppressants, and/or anti-TNF-agents). The main objective in this cohort was to determine whether catalase regulation is restored when patients enter the remission phase; all experiments were performed again on these individuals. The rest of the patients displayed an aggressive development of the disease that made it difficult to confirm remission. The mucosal healing was evaluated by magnetic resonance enterography or ileocolonoscopy, according to disease localization and scored according to the quantitative Magnetic Resonance Index of Activity (MaRIA) [71] and the Crohn’s Disease Endoscopic Index of Severity (CDEIS) respectively [72].

All enrolled subjects gave their informed written consent, and upon recruitment, a complete hematological and biochemistry blood analysis for each subject was performed, including acute phase reactants, such as C-Reactive protein (CRP), fibrinogen, and erythrocyte sedimentation rate (ESR). Only those with a completely normal blood test result were included as the controls. The study was conducted according to the tenets of the Declaration of Helsinki and was approved by the ethical committee of the Hospital Universitari La Fe (no. PI14/01702).

### 4.2. Single Nucleotide Polymorphisms Analysis

The selection of *CAT* SNPs was based on those previously described in the literature for having a role in other related diseases in which the inhibition of CAT was reported [42,43,44,45,46]. The SNPs analyzed were rs1001179, rs12273124, rs17886155, rs2268064, rs2284365, rs3758730, rs475043, rs494024, rs525938, rs564250, rs704724, rs769217, and rs7943316. Briefly, peripheral blood samples (6 mL) were collected in EDTA-containing tubes and centrifuged at 3500 rpm for 30 min without a break. White blood cells were then isolated and washed with PBS. Following centrifugation at 1500 rpm for 15 min, the resulting supernatant was discarded, and the leukocyte pellet was collected. After the isolation of the nucleated cells, genomic DNA was extracted using the Chemagic DNA Blood Kit on a Chemagen Module I workstation (Chemagen Biopolymer-Technologie AG, Baesweiler, Germany), according to the manufacturer’s instructions. Genotyping for the polymorphisms was carried out using the Agena Biosciences MassArray platform, following the manufacturer’s protocol (Agena Biosciences, San Diego, CA, USA). The SNP assay was designed with Assay Design Suite (https://www.agenabio.com/?s=assay+design, accessed on 1 December 2022) using the default settings. PCR primers were pooled to a final concentration of 500 nM and used to amplify 10 ng of DNA in a 5 µL volume reaction with 1U of FastStart Taq Polymerase (Roche, Indianapolis, IN, USA) and 4 mM magnesium chloride. The polymerase chain reactions were carried out in 5 µL volumes in a standard 384-well plate formatted according to the specifications provided by Agena Biosciences. The amplified product was cleaned using shrimp alkaline phosphatase to neutralize any unincorporated dNTP. Allele discrimination reactions were conducted by adding extension primers, DNA polymerase, and a cocktail of dNTPs and ddNTPs to each well. The reaction mixture was then spotted on a SpectroCHIP II microarray and subjected to MALDI-TOF mass spectrometry, according to the iPLEX Gold Application Guide [73].

### 4.3. RNA Isolation and Gene Expression Analysis

Whole blood samples (9–10 mL) in EDTA-containing tubes were collected and immediately processed. Total RNA from the leukocyte population was extracted according to the manufacturer’s protocol (*LeukoLOCK™ Total RNA Isolation System*; Ambion, Austin, TX, USA), and 1000 ng was used for reverse transcription. cDNA synthesis was performed using the *High-Capacity RNA-to-cDNA Kit* (Applied Biosystems Pty Ltd., Scoresby, Australia). The obtained cDNA was diluted 1/10 in water and mixed with a synthetic competitor (with an SNP mismatch) for the realization of a competitive PCR using the *Agena Biosciences´s MassARRAY^®®^ quantitative gene expression (QGE) analysis application* (Agena Biosciences^TM^, San Diego, CA, USA) in duplicate. Data of mRNA expression was analyzed using *MassARRAY QGE Analyzer v3.4* (Agena Biosciences, San Diego, CA, USA) with copy numbers for each transcript per sample determined based on the EC50 of standard curve titrations of known competitor amounts per assay versus a fixed amount of cDNA template. Primers and competitors were designed using the *QGE Assay Design–Multiplexed QGE Assay Design software* (Agena Biosciences^TM^, San Diego, CA, USA) containing identical tags at their 5′ends (sequences are detailed in the Appendix A).

### 4.4. Isolation of Peripheral Leukocytes

Anticoagulated blood (K3-EDTA) was obtained from patients and healthy volunteers after 12 h of fasting. Leukocytes were isolated by the Histopaque 1077 gravity technique and both leukocytes and plasma were stored separately at −80 °C until use.

### 4.5. Western Blotting

The protein concentration of the CAT was measured by Western blot using specific monoclonal antibody. To remove red blood cell contamination, the leukocyte pellet was washed three times with 0.9% NaCl solution and subsequently haemolysed by the addition of distilled water. The cell pellets were subsequently lysed with a lysis buffer containing HEPES [pH 7.4], Na Cl, NaF, b-glicerophosphate, PMSF, Triton, phosphatase inhibitor, and protease inhibitor. The samples were centrifuged at 13,000 g for 10 min at 4 °C, and the supernatant was collected. Protein concentrations in leukocyte lysates were determined using the method of Lowry et al. [74]. Equal amounts of protein (20 μg of sample in a total volume 30 μL) were separated on a 12% reducing SDS-polyacrylamide gel, transferred onto a Protran nitrocellulose membrane (Schleicher and Schuell Bio-Science, Keene, NH, USA), blocked with TBST (Tris-buffered saline [pH 7.6] and 0.05% Tween 20), and supplemented with 5% nonfat milk. Membranes were incubated overnight with the monoclonal antibody anti-CAT 505 at 1:7000 dilutions (Sigma, Merck Life Science S.L. Madrid, Spain; C0979). Anti-actin (Sigma; A2066) was used as a loading control. Membranes were incubated with different secondary antibodies according to the primary antibody for 45 min. Anti-Mouse IgG Peroxidase conjugated (Calbiochem, Merck Millipore. Madrid. Spain; 401215) was used for CAT. The signals were quantified with a scanning densitometer (Fujifilm LAS-1000 plus). The 66-kD protein was identified as CAT and the 43-kD protein as actin.

### 4.6. Protein Kinase C Zeta (PKCζ) and Protein Phosphatase 2A (PP2A) Levels

The protein levels of the catalase regulators PKCζ and PPA2 in the peripheral blood cells from the confirmatory cohort were determined using human PKCζ (Ref. CSB-EL018710HU; CUSABIO^®®^) and human PPA2 (Ref. CSB-EL018411HU; CUSABIO^®®^) kits, according to manufacturer’s instructions. Previously, preliminary experiments to identify the optimal sample quantity and dilution were performed. Then, the pellets of the peripheral blood cells were analyzed following mainly the manufacturer’s instructions, including homogenization by mechanical disruption of cell membranes by freeze-thaw cycles, sonication bathing, and centrifugation to eliminate undesirable particles. Supernatants were also analyzed by BCA method (Pierce^®®^) to determine total protein concentration to normalize the results.

### 4.7. Statistical Analysis

Data from the SNP assays were summarized using mean (standard deviation) and median (1st and 3rd Q) in the case of the continuous variables and with relative and absolute frequencies in the case of the categorical variables. A Pearson’s chi-square test was employed to study the association between SNPs in the *CAT*-gene of healthy subjects and patients. Three genetic models (recessive, log-additive, and codominant) were used to statistically quantify the genetic association of the studied SNPs to the frequency of CD, adjusted for covariates (age, smoking, sex, disease location, and disease behavior). A Bonferroni test was used as a statistical correcting method. The association between the different SNPs and CD was assessed using penalized logistic regression with the elastic net algorithm. The penalization parameter was selected using 500 repetitions of 10-fold cross-validation. The SNPs selected by the elastic net algorithm were represented using Cohen-Friendly association plots. All statistical analyses were performed using R (version 3.5.2) and the R package glmnet (version 2.0–16).

Demographic and clinical data are expressed as mean ± SD for the quantitative variables and as numbers (percentages) for the categorical variables. Data from the CAT functional and regulatory experiments are shown as mean ± SEM. The one-sample Kolmogorov–Smirnov test was used to verify the normal distribution of the data. Statistical significance was tested by Student’s *t*-test (two-tailed, 95% confidence interval) or Mann–Whitney U test. A *p*-value < 0.05 was considered significant in all analyses. All these statistical analyses were carried out using GraphPad software version 8.0 (GraphPad Software Inc., San Diego, CA, USA).

## 5. Conclusions

Our study has shown that the permanent reduction in the CAT enzyme observed in CD patients may be due to genetic changes, highlighting *CAT* SNPs rs475043, which may modulate CD susceptibility and the phenotype of CD. We also observed a deficiency in gene and protein CAT expression, as well as the dysregulation of the main CAT regulators, PP2A and PKC. The inhibition of this antioxidant at different levels may contribute to the pathophysiology of CD.

## Figures and Tables

**Figure 1 ijms-23-15881-f001:**
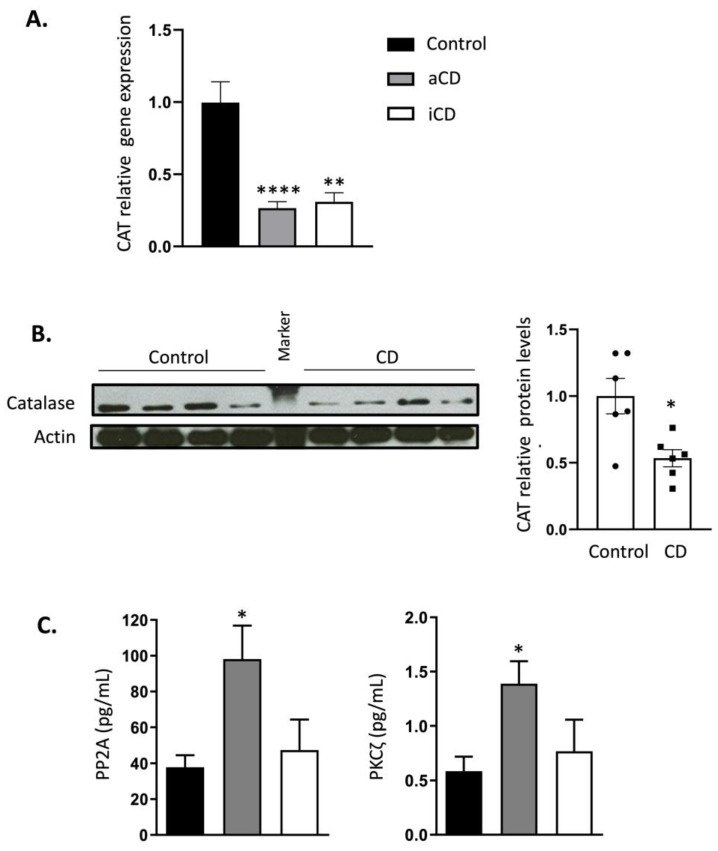
Permanent CAT inhibition in leukocytes isolated from Crohn’s disease (CD) patients. (**A**). Gene expression analysis of CAT in the controls and active and inactive CD patients (*N* = 18, 20, and 10, respectively). (**B**). A Western blot densitometer, normalized with α-actin; its quantification represented in a histogram. The protein levels of CAT in the controls were significantly higher than in CD patients (*N* = 6 and 6, respectively) normalized with α-actin. (**C**). The protein levels (pg/mL) of CAT regulators Protein Phosphatase 2A (PP2A) and Protein Kinase C zeta (PKCζ) in the controls; active and inactive CD patients (*N* = 7, 10, and 4, respectively). The levels of both regulators had increased more in CD than in the control and iCD patients. The results are shown as mean ± SEM. The differences between the controls and both the active and inactive CD patients (aCD and iCD, respectively) were assessed by Student’s *t*-test. * *p* < 0.05, ** *p* < 0.01, **** *p* < 0.0001.

**Table 1 ijms-23-15881-t001:** Demographic characteristics and clinical data of patients and matched controls.

		Control (*n* = 625)	CD (*n* = 598)
Age		47.34 ± 8.54	42.06 ± 15.17
Gender (M/F)		389/236	281/317
Montreal Classification			
A	A1:		30 (5%)
	A2:		401 (67%)
	A3:		167 (28%)
L	L1:		227 (38%)
	L2:		113 (19%)
	L3:		251 (42%)
	L4:		28 (4%)
B	B1:		376 (63%)
	B2:		132 (22%)
	B3:		90 (15%)
Perianal disease			148 (24%)
Treatments:			363 (82%)
Immunossupressors			371 (62%)
Biologic therapy			
Family history			82 (15%)
Extraintestinal manifestations			119 (20%)
Surgeries			271 (45%)
Smokers			185 (31%)
Nonsmokers			269 (45%)
Exsmokers			144 (24%)

Abbreviations: CD, Crohn’s disease; A: Age at diagnosis: A1 ≤ 16 years; A2 17–40 years; A3 > 40; L: Location: L1 = ileal; L2 = colonic; L3 =ileocolonic; B: Behavior: B1 = nonstricturing, nonpenetrating Crohn’s disease; B2 = structuring Crohn’s disease; B3 = penetrating Crohn’s disease.

**Table 2 ijms-23-15881-t002:** Genotype association analysis of *CAT* SNPs with clinical variables of discovery cohort.

CAT SNPs	Genotype	Significant Clinical Associations
rs1001179	CC	Protect perianal disease
rs12273124	AG	Increase inflammatory behavior in smokers
rs3758730	TA	Protect stricturing behavior and ileocolonic location
	TT	Increase stricturing behavior in smokers
rs475043	AG	Increase upper digestive location and penetrating behavior in smokers
	GG	Protect upper digestive location
rs494024	GA	Protect penetrating behavior
	GG	Protect inflammatory behavior
rs564250	TC	Increase inflammatory behavior in smokers
	TT	Increase penetrating behavior

*N* = 625 for controls and 598 for CD patients. The association of the different SNPs with CD was assessed using penalized logistic regression with the elastic net algorithm.

**Table 3 ijms-23-15881-t003:** Association of single nucleotide polymorphisms in the *CAT*-gene of peripheral blood samples of healthy subjects (controls) and Crohn’s disease (patient).

rs7943316	Control	Patient	*p* Value
AA	10	7	0.04 *
AT	55	61	
TT	35	32	
rs3758730			
AA	76.5	76.5	0.23
TA	22	23	
TT	1.5	0.5	
rs475043			
AA	45	40	0.036 *
AG	46	53	
GG	9	7	
rs769217			
CC	45	51	0.041 *
CT	49	45	
TT	6	4	
rs525938			
AA	38	46	0.007 *
AG	53	48	
GG	9	6	
rs564250			
CC	51	57	0.089
CT	46	41	
TT	3	2	
rs1001179			
AA	3	2	0.27
GA	36	36	
GG	31	62	

Data are shown as absolute numbers (allele distribution in percentage). *N* = 625 for controls and 598 for CD patients. * *p* < 0.05

**Table 4 ijms-23-15881-t004:** Demographic and disease characteristics of the confirmatory human cohort. Controls and CD patients at baseline (active CD) and after achieving disease remission (inactive CD).

KERRYPNX	Control (*n* = 18)	aCD (*n* = 20)	iCD (*n* = 10)
Age	27.83 ± 5.24	32.04 ± 9.14	32.1 ± 10.1
Gender (M/F)	4/14	7/13	6/4
Harvey Index	--	7,31 ± 2,86	<4
CRP	2.8 (2)	70.31 ± 95.24 ^a,b^	3 ± 3.97
Fibrinogen	287 ± 69	531.42 ± 129.18 ^a^	349 ± 81.4
ESR	11.8 ± 12	57.87 ± 31.6 ^a,b^	10.25 ± 10.27
S/nS/eS	0/14/2	11/8/1	2/7/1
Montreal Classification			
A			
A1:	-	0 (0%)	0 (0%)
A2:	-	20 (100%)	10 (100%)
A3:	-	0 (0%)	0 (0%)
L			
L1:	-	13 (65%)	7 (70%)
L2:	-	3 (15%)	3 (30%)
L3:	-	4 (20%)	0 (0%)
L4:	-	0 (0%)	0 (0%)
B			
B1	-	18 (90%)	8 (80%)
B2:	-	2 (10%)	2 (20%)
B3:	-	0 (0%)	0 (0%)
*p*			
Yes:	-	4 (20%)	0 (0%)
No:	-	16 (80%)	10 (100%)
No:			
Time until remission (months)			14.87 ± 10.9

Abbreviations: A, Age at diagnosis: A1 ≤ 16 years; A2 17–39 years; A3 ≥ 40; L, Location: L1 = ileal; L2 = colonic; L3 = ileocolonic; B, Behavior: B1 = non-stricturing, non-penetrating Crohn’s disease; B2 = stricturing Crohn’s disease; B3 = penetrating Crohn’s disease; p, perianal disease; CRP, C-reactive protein; ESR, erythrocyte sedimentation rate; S/nS/eS, smokers/nonsmokers/exsmokers; aCD, active Crohn’s disease; iCD, inactive Crohn’s disease. Values are given as the mean ± SD or as absolute numbers (percentage) where applicable. ^a^
*p* <0.05 significant differences compared with the control group. ^b^
*p* < 0.05 significant differences compared inactive CD group.

## Data Availability

Any data or material that support the findings of this study can be made available by the corresponding author upon request.

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
