# Peer review of "The Genetic Diversity and Dysfunctionality of Catalase Associated with a Worse Outcome in Crohn’s Disease"

_ijms, 2022, doi:10.3390/ijms232415881_

Round 1
Reviewer 1 Report
The authors present a genetic study of catalase diversity and dysfunction associated with worse outcome in Crohn's disease.
Their study is based on the fact that chronic inflammation of the intestine in Crohn's disease (CD) is associated with an increase in oxidative stress and imbalance of antioxidant enzymes.
ü The authors should check acronyms like IBD at line 104 and specify, at least the first time in full, what they correspond to.
ü Line 141 the authors write:” CAT expression values were normalized with GPDH, HET2 and TNFR and those from protein levels were normalized with α-actin” Did the authors consider that GPDH and α-actin may vary with oxidative stress and therefore they are not good housekeeping genes to use as normalizers?(Sohn JY, Kwak HJ, Rhim JH, Yeo EJ. AMP-activated protein kinase-dependent nuclear localization of glyceraldehyde 3-phosphate dehydrogenase in senescent human diploid fibroblasts. Aging (Albany NY). (2022)
To choose an adequate housekeeping, the authors could consult the paper “Succoio M, Comegna M, D'Ambrosio C, Scaloni A, Cimino F, Faraonio R. Proteomic analysis reveals novel common genes modulated in both replicative and stress-induced senescence. J Proteomics. 2015”, in which oxidative stress was induced to determine premature senescence.
ü The authors should check Figure 1 and better describe in the legend what panels represent by specifying each graph. For example, in Figure 1(B) there are two images (a western blot and a histogram) what do they stand for?
ü The authors should check if the “rs” number is right and if they correspond to what is specified.
For example, they should verify what they write from lines 274 to 277. In particular, they write “Genomic DNA was analyzed for 14 SNPs of the CAT gene (rs1001179, rs1141718, rs12273124, rs17886155, rs2268064, rs2284365, rs3758730, rs475043, rs494024, rs4987023, rs525938, rs564250, rs5746129, rs704724, rs769217 and rs7943316).” But the “rs” are 16 and they are not only of the CAT gene but there are 3 SNPs of the SOD2 gene.
ü In the table 1, the authors specify the ages of the subjects enrolled in the study. It is evident that there is a group (A3) > 40 years. There is a vast literature on the relationship between aging and oxidative stress. Did the authors consider this aspect and any additional effect?
ü Line 298, the authors affirm that the enrolled subjects have signed an informed consent but among them there are also minors therefore it would be appropriate to specify “legal guardians for minor” if it has been done.
ü “4.3. RNA isolation and gene expression analysis”, did authors make experiments in duplicate/triplicate?
ü “4.5. Western Blotting”, what is the extraction method for proteins?
ü Line 351, authors write “Equal amounts of protein were separated on a 12% reducing SDS-poly- 351 acrylamide gel” , how much mg were used exactly?
Author Response
November 2022
Dear reviewers,
Sincerest thanks for your response. The authors thank the Editors and Reviewers for their helpful recommendations. The inputs of the reviewers have enabled us to further improve the quality of our manuscript.
This letter includes a detailed response to each of the comments and a description of changes made to the manuscript. We also include a copy of the manuscript with all the changes highlighted in red.
We shall look forward to hearing from you at your earliest convenience.
Yours sincerely,
Dr Marisa Iborra
REVIEW 1 Comments and Suggestions for Authors
The authors present a genetic study of catalase diversity and dysfunction associated with worse outcome in Crohn's disease.
Their study is based on the fact that chronic inflammation of the intestine in Crohn's disease (CD) is associated with an increase in oxidative stress and imbalance of antioxidant enzymes.
ü The authors should check acronyms like IBD at line 104 and specify, at least the first time in full, what they correspond to.
Reply: We have changed it.
ü Line 141 the authors write:” CAT expression values were normalized with GPDH, HET2 and TNFR and those from protein levels were normalized with α-actin” Did the authors consider that GPDH and α-actin may vary with oxidative stress and therefore they are not good housekeeping genes to use as normalizers?(Sohn JY, Kwak HJ, Rhim JH, Yeo EJ. AMP-activated protein kinase-dependent nuclear localization of glyceraldehyde 3-phosphate dehydrogenase in senescent human diploid fibroblasts. Aging (Albany NY). (2022)
To choose an adequate housekeeping, the authors could consult the paper “Succoio M, Comegna M, D'Ambrosio C, Scaloni A, Cimino F, Faraonio R. Proteomic analysis reveals novel common genes modulated in both replicative and stress-induced senescence. J Proteomics. 2015”, in which oxidative stress was induced to determine premature senescence.
We thank the reviewer for the comment. At the time of the experimental analysis, we selected the normalizers according to the information described in the scientific literature
*Ibañez-Cabellos JS, Seco-Cervera M, Perez-Machado G, Garcia-Gimenez JL, Pallardo FV. Characterization of the antioxidant systems in different complementation groups of Dyskeratosis Congenita. Free Radic Biol Med. 2014
*Zapatero-Solana E, García-Giménez JL, Guerrero-Aspizua S, García M, Toll A, Baselga E, Durán-Moreno M, Markovic J, García-Verdugo JM, Conti CJ, Has C, Larcher F, Pallardó FV, Del Rio M. Oxidative stress and mitochondrial dysfunction in Kindler syndrome. Orphanet J Rare Dis. 2014).
The papers mentioned by the reviewer are dater later of our experiments.
We agree with the reviewer because it is a mistake: Normalization of copy numbers between samples for the different assays was conducted using a multiplexed set of 3 well-characterized human housekeeping (internal controls) transcripts (ACTB-6, GAPDH-6 AND HMBS-7) and geNorm software.
* Iborra M, Moret I, Rausell F, Busó E, Cerrillo E, Sáez-González E, Nos P, Beltrán B. Different Genetic Expression Profiles of Oxidative Stress and Apoptosis-Related Genes in Crohn's Disease. Digestion. 2019.
We have corrected the housekeepers used.
ü The authors should check Figure 1 and better describe in the legend what panels represent by specifying each graph. For example, in Figure 1(B) there are two images (a western blot and a histogram) what do they stand for?
We have added an explanation of the figure in order to improve the understanding.
ü The authors should check if the “rs” number is right and if they correspond to what is specified.
For example, they should verify what they write from lines 274 to 277. In particular, they write “Genomic DNA was analyzed for 14 SNPs of the CAT gene (rs1001179, rs1141718, rs12273124, rs17886155, rs2268064, rs2284365, rs3758730, rs475043, rs494024, rs4987023, rs525938, rs564250, rs5746129, rs704724, rs769217 and rs7943316).” But the “rs” are 16 and they are not only of the CAT gene but there are 3 SNPs of the SOD2 gene.
Thank you very much for the observation. We have revised the paper and following the reviewer’s suggestions. The new sentence in material and methods is as follow ” Blood samples were obtained from all the participants. Genomic DNA was analyzed for 13 SNPs of the CAT gene (rs1001179, rs12273124, rs17886155, rs2268064, rs2284365, rs3758730, rs475043, rs494024, rs525938, rs564250, rs704724, rs769217 and rs7943316).
ü In the table 1, the authors specify the ages of the subjects enrolled in the study. It is evident that there is a group (A3) > 40 years. There is a vast literature on the relationship between aging and oxidative stress. Did the authors consider this aspect and any additional effect?
It is a good consideration. We had not contemplated this aspect in the present study because the effect of the aging was not the main aim. We only considered the potential effect of the employed treatments (immunosuppressors). The involved cohort did not show significant differences in the age (controls 47.34 ± 8.54 vs CD 42.06 ± 15.17 and controls 27,83 ± 5,24 vs CD 32,04 ± 9,14 in confirmatory cohort). However, future studies could be development in order to explain if there is a relationship.
ü Line 298, the authors affirm that the enrolled subjects have signed an informed consent but among them there are also minors therefore it would be appropriate to specify “legal guardians for minor” if it has been done.
Certainly, we have added this consideration into the text.
ü “4.3. RNA isolation and gene expression analysis”, did authors make experiments in duplicate/triplicate?
We acknowledge your remark. The analysis were made in duplicate, and we have added this information into the text.
ü “4.5. Western Blotting”, what is the extraction method for proteins?
Initially, we remove this part of the text in order to shorten the length of the paper. According with the reviewer, we add the next paragraph:
“The protein concentration of the CAT was measured by western blot, using specific monoclonal antibody. To remove red blood cells contamination, the leukocyte pellet was washed three times with 0.9% NaCl solution and subsequently haemolysed by the addition of distilled water. The cell pellets were subsequently lysed with a lysis buffer containing HEPES [pH 7.4], Na Cl, NaF, b-glicerophosphate, PMSF, Triton, phosphatase inhibitor and protease inhibitor. The samples were centrifuged at 13000 g for 10 min at 4ºC and the supernatant was collected.”
ü Line 351, authors write “Equal amounts of protein were separated on a 12% reducing SDS-poly- 351 acrylamide gel” , how much mg were used exactly?
We used 20 μg of sample in a total volume 30 μl. Following your suggestion, we have added this information into the text.
Reviewer 2 Report
Abstract is too long, please shorten according to guidelines for authors.
Lines 103-104: I think those publications mention only rs475043. Try to rephrase the sentence.
Table 1: Montreal Classification for disease localization does not sum up to 100%. Explain.
Table 2: Did you use any statistical correcting methods? Consider using Bonferroni corrections.
Consider changing to : Table 3. Association of Single Nucleotide Polymorphisms in the CAT-gene of peripheral blood samples of healthy subjects (control) and Crohn’s disease (patient)
Does the asterisks means statistically significant result? It is not clear. Define asterisk in footnote to the table
Usually in the publication, tables are without lines between rows. Check the instructions for authors of the journal
The Table 3 would be much clearer if you put the genotypes in the first column in three separated rows rather than in separate columns.
What is the rationale to use Pearson’s p in Table 3? What kind of statistical test was used to compare the distributions? Please provide this information in the methods section. I think two tailed Fisher’s exact test would be better in here. Please provide odds ratios with 95% confidence interval for each comparison in Table 3.
Line 119: There is no mention of CRP, fibrinogen, and ESR measurments in the Method sections. Also please define the these abbreviations when first used (not only in the Table footnotes)
Table 4: Consider defining Montreal Classification abbreviations one more time in the table’s footnotes.
Line 153: change Chron to Crohn, check through the whole text
Line 159: the *** is missing, change to make it consistent
Figure 1: I suggest writing the number of patients under each graph. Try to include the legend also for the panel C.
Line 318: the link appears to be broken
Line 398: consider using word “pathophysiology” as it is more commonly used according to google trends
Author Response
November 2022
Dear reviewer,
Sincerest thanks for your response. The authors thank the Editors and Reviewers for their helpful recommendations. The inputs of the reviewers have enabled us to further improve the quality of our manuscript.
This letter includes a detailed response to each of the comments and a description of changes made to the manuscript. We also include a copy of the manuscript with all the changes highlighted in red.
We shall look forward to hearing from you at your earliest convenience.
Yours sincerely,
Dr Marisa Iborra
REVIEW 2 Comments and Suggestions for Authors
- Abstract is too long, please shorten according to guidelines for authors.
According to the rules of the editorial, we have reduced the abstract.
- Lines 103-104: I think those publications mention only rs475043. Try to rephrase the sentence.
Thank for the observation. We have included the reference 16 to mention all the SNP.
- Table 1: Montreal Classification for disease localization does not sum up to 100%. Explain.
Montreal Classification defines L4 as any disease location proximal to the terminal ileum. L4 is a modifier that can be added to L1-L3 (ileal, colonic or ileocolonic location), when concomitant upper gastrointestinal disease is present.
- Table 2: Did you use any statistical correcting methods? Consider using Bonferroni corrections.
Yes, we have used Bonferroni correction test. We have added this clarification in the test.
- Consider changing to : Table 3. Association of Single Nucleotide Polymorphisms in the CAT-gene of peripheral blood samples of healthy subjects (control) and Crohn’s disease (patient)
We have change it.
- Does the asterisks means statistically significant result? It is not clear. Define asterisk in footnote to the table
We have defined asterisk in footnote of the table such statistically significant (p<0.05).
- Usually in the publication, tables are without lines between rows. Check the instructions for authors of the journal
We have removed the lines.
- The Table 3 would be much clearer if you put the genotypes in the first column in three separated rows rather than in separate columns.
Accordingly, we have changed the table 3 with the reviewer instructions.
-What is the rationale to use Pearson’s p in Table 3? What kind of statistical test was used to compare the distributions? Please provide this information in the methods section. I think two tailed Fisher’s exact test would be better in here. Please provide odds ratios with 95% confidence interval for each comparison in Table 3.
Thank for the comment and sorry for the mistake. P refers to the p-value obtained from Pearson's chi-square test. We have used this statistical test applied to sets of categorical data to evaluate how likely it is that any observed difference between the sets arose by chance. We believe that confidence interval is not suitable for qualitative variables.
We have added this comment in the text and have modified the fault.
-Line 119: There is no mention of CRP, fibrinogen, and ESR measurments in the Method sections. Also please define the these abbreviations when first used (not only in the Table footnotes)
We added the parameters to the sentence: “a complete hematological and biochemistry blood analysis for each subject was performed, including acute phase reactants such as C-Reactive protein (CRP), fibrinogen and erythrocyte sedimentation rate (ESR).
-Table 4: Consider defining Montreal Classification abbreviations one more time in the table’s footnotes.
We have added the Montreal Classifications abbrevations in the table´s footnotes.
- Line 153: change Chron to Crohn, check through the whole text
We have changed it.
-Line 159: the *** is missing, change to make it consistent
We have changed it.
- Figure 1: I suggest writing the number of patients under each graph. Try to include the legend also for the panel C.
We have included the number of patients in the legend: C. Protein levels (pg/mL) of CAT regulators PP2A and PKCζ in controls, active and inactive CD patients (N=7, 10 and 4, respectively). The levels of both regulators were more increased in aCD than in control and iCD patients.
-Line 318: the link appears to be broken
Thank you, the link has been updated.
- Line 398: consider using word “pathophysiology” as it is more commonly used according to google trends
We have modified it.

Round 2
Reviewer 2 Report
Please make sure that the use of italics and the number of decimal places in Table 3 are correct.
I strongly recommend adding a legend in Figure 1 C.
Author Response
Dear Reviewer,
Thank for the remarks and sorry for the inconvenience. Next, we answer your comments.
- Please make sure that the use of italics and the number of decimal places in Table 3 are correct.
Thank you for your comment. We have changed the use of italic and shortened the number of decimal places (only 3).
- I strongly recommend adding a legend in Figure 1 C.
We have added the legend: "C. Protein levels (pg/mL) of CAT regulators Protein Phosphatase 2A (PP2A) and Protein Kinase C zeta (PKCζ) in controls, active and inactive CD patients (N=7, 10 and 4, respectively). The levels of both regulators were more increased in aCD than in control and iCD patients."